# Forecasting Bitcoin Trends Using Algorithmic Learning Systems

**DOI:** 10.3390/e22080838

**Published:** 2020-07-30

**Authors:** Gil Cohen

**Affiliations:** Department of Management, Western Galilee Academic College, P.O.Box, 2125, Acre 2412101, Israel; gilc@wgalil.ac.il

**Keywords:** Bitcoin, algorithmic trading, Darvas box, swarm optimizations

## Abstract

This research has examined the ability of two forecasting methods to forecast Bitcoin’s price trends. The research is based on Bitcoin—USA dollar prices from the beginning of 2012 until the end of March 2020. Such a long period of time that includes volatile periods with strong up and downtrends introduces challenges to any forecasting system. We use particle swarm optimization to find the best forecasting combinations of setups. Results show that Bitcoin’s price changes do not follow the “Random Walk” efficient market hypothesis and that both Darvas Box and Linear Regression techniques can help traders to predict the bitcoin’s price trends. We also find that both methodologies work better predicting an uptrend than a downtrend. The best setup for the Darvas Box strategy is six days of formation. A Darvas box uptrend signal was found efficient predicting four sequential daily returns while a downtrend signal faded after two days on average. The best setup for the Linear Regression model is 42 days with 1 standard deviation.

## 1. Introduction

Because Bitcoin’s prices involve high volatility (for example: Baek and Elbeck [1]), traders may benefit from using short-term trading systems. There is a lack of research that has tried to adjust and optimize such systems to Bitcoin’s price changes. Gharehchopogh et al. [2] proved that linear Regression can predict S&P 500 volumes of trade. Tiong et al. [3] utilized linear regression equation to analyze and discover the trends patterns in forex markets and found that this approach has been proven beneficial. Munim et al. [4] forecast next-day Bitcoin price using the autoregressive integrated moving average (ARIMA) and neural network autoregression (NNAR) models. The Diebold Mariano test they conducted, confirmed the superiority of forecast results of ARIMA model over NNAR in the test-sample periods. Their paper demonstrates that ARIMA model enduring power of volatile Bitcoin price prediction. Atsalakis et al. [5] proposed a computational intelligence technique that uses a hybrid neuro-fuzzy controller to forecast the direction in the change of the daily price of Bitcoin. The proposed methodology outperforms two other computational intelligence models, the first being developed with a simpler neuro-fuzzy approach, and the second being developed with artificial neural networks. Furthermore, the investment returns achieved by a trading simulation, based on the signals of the proposed model, are 71.21% higher than the ones achieved through a naïve buy-and-hold strategy. As did Atsalakis et al. [5], we also compare our trading systems results to a simple buy-and-hold strategy.

Since cryptocurrencies are relatively new, there is little research on the factors or tools that can help people invest in them. Moore and Christin [6] have studied the risk involved in Bitcoin exchanges, which convert Bitcoin to hard currency and vice versa. They found that the exchange’s transaction volume is a good proxy as to whether it is likely to close. Nevertheless, while less popular exchanges are more likely to be shut down, popular exchanges are more likely to suffer security breaches.

Trading strategies have been developed in the past by various researchers. A popular approach was to face trading as an optimal stopping problem (for example, Chow et al. [7]). Liu et al. [8] modeled the “buy low sell high” trading practice for currencies pairs, proving that this strategy is profitable while the only risk comes from a wrongly predicted positive drift when the market plunges. A group of studies linked the price of Bitcoin to social networks. Kim et al. [9] tried to predict fluctuations in the prices of cryptocurrencies by analyzing comments in online cryptocurrency communities. They found that positive user comments significantly affected the price fluctuations of Bitcoin, whereas the prices of two other cryptocurrencies, Ripple and Ethereum were strongly influenced by negative user comments and replies. Garcia and Schweizer [10] demonstrated the existence of a relationship between Bitcoin’s returns and Twitter’s valence and polarization information. Matta et al. [11] studied the existing relationship between Bitcoin’s trading volumes and the number of queries on Google. They reported significant cross correlation values, demonstrating that the volume of searches could predict the trading volume of Bitcoin. Wołk [12] concluded that cryptocurrency price fluctuations depend heavily on social media sentiment and web search analytics tools such as Google Trends. Twitter sentiments regarding future cryptocurrency prices tend to be positive as many people tweet about cryptocurrencies even if their prices go down. However, it is difficult to predict cryptocurrency prices because of their volatile nature in the current market. They found a combination of Google trends data and general negative sentiments to be the most powerful predictors. Moreover, they conducted an experiment by trading $100 on the BitBay cryptocurrency exchange for one month (December 2018). After one month, their account balance stood at $114.82, when the buy-and-hold strategy would have resulted ended the month with $92 using bitcoin, $117 using Ethereum, $94 using Ripple (XRP), and $71 using ZEC. In contrast to Wołk [12], our research depends on a long run investment approach using three years of strategy forming and more than five years of simulated trading for both Darvas Box and Linear Regression models. Abraham et al. [13] predicted changes in Bitcoin and Ethereum prices utilizing Twitter data and Google Trends data. They claim that Twitter is increasingly used as a news source of information that influences purchasing decisions. By analyzing tweets, they found that tweet volume, rather than tweet sentiment, is a good predictor of price direction. By utilizing a linear model that uses tweets and Google Trends data, they were able to accurately predict the direction of price changes. Jay et al. [14] proposed a stochastic neural network model for cryptocurrency price prediction. The proposed that the approach is based on the random walk theory, which is widely used in financial markets for modeling stock prices. The proposed model induces layer-wise randomness into the observed feature activations of neural networks to simulate market volatility. They trained the multi-layer perceptron and long short-term memory models for Bitcoin, Ethereum, and Litecoin. The results show that the proposed model is superior in comparison to the deterministic models. Almost all the stochastic versions of the neural net models outperformed the deterministic versions. The average relative improvement by using stochastic neural networks over regular neural networks was 1.56%.

Another group of studies examined Bitcoin’s market inefficiencies. Balcilar et al. [15] discussed the predictability of Bitcoin’s returns and volatility based on transaction volume. They found that when extreme events are excluded, volume is an important predictor of price. In studying Bitcoin’s price dynamics and trading, Blau [16] concluded that speculative behavior could not be directly linked to the unusual volatility of the Bitcoin market. Brandvold et al. [17] investigated the role of various Bitcoin exchanges in the price discovery process, noting that the information share is dynamic and evolves significantly over time. Feng et al. [18] found evidence of informed trading in the Bitcoin market prior to major events. Moreover, when examining the timing of informed trades, they noticed that informed traders prefer to build their positions two days before large positive events and one day before large negative events. This result serves as proof of the market inefficiency that differentiates uninformed traders from informed traders of Bitcoin. Caporale and Plastun [19] examined the day of the week effect in the cryptocurrency market. They determined that most cryptocurrencies such as Litecoin, Ripple and Dash do not exhibit this anomaly. The only exception is Bitcoin, for which returns on Mondays are significantly higher than those on the other days of the week.

As described above, Bitcoin research concentrates mainly on two factors: the effect of social media on cryptocurrency prices and market anomalies. The effectiveness of Darvas boxes and Linear Regression techniques for Bitcoin trend predictions and trading has never been tested in prior research. In the following research, we used Support Vector Machine (SVM) to predict Bitcoin price trends. SVM is a technique of supervised learning that enable a machine to learn and aggregate knowledge from training data and produce an inferred function which can be used for mapping new examples and predict future events. Each example consists of an input objects and a desired output value that serves as a supervisory signal. The training examples are usually represented as vectors in a linear spectrum. A classifying factor should enable the machine to differentiate between positive and negative training examples. A well-known SVM procedure is called classification techniques that focus on predicting a qualitative response by analyzing data and recognizing patterns.

The patterned that was applied to past Bitcoin prices is the famous Darvas box pattern that is commonly used by stocks and commodities analysts. Nicolas Darvas developed his theory in the 1950s while travelling the world as a professional ballroom dancer. His trading technique involves drawing a box around the recent highs and lows of the financial asset to establish entry point and placement of the stop-loss order. The theory uses market momentum theory along with technical analysis to determine when to enter and exit the market. Darvas believed his method worked best when applied to industries with the greatest potential to excite investors and consumers with revolutionary products. The fact that Bitcoin is a relatively new and revolutionary in terms of a digital currency without a central bank or a single other administrator or intermediaries, has fascinated investors around the globe. Those characteristics makes it a perfect candidate for price trends forecasting using Darvas Boxes technique.

Since its introduction in 2009, Bitcoin’s value has risen and fallen with a volatility that has made it difficult for investors to achieve positive gains on their investment. As a result, investors have sought algorithms to help them improve their investment forecasting ability. When developing the system, Darvas trading techniques has been used frequently on growth and volatile stocks. However, it has never been tried on Bitcoin price predictions. Linear regression is a supervised learning technique typically used in predicting, forecasting, and finding relationships between quantitative data. It is one of the earliest learning techniques, which is still widely used. A Linear Regression line is a straight line that best fits the prices between a starting price point and an ending price point. A “best fit” means that a line is constructed where there is the least amount of space between the price points and the actual Linear Regression line. When prices are below the Linear Regression Line, this could be viewed by some traders as a good time to buy, and when prices are above the Linear Regression line, a trader might sell. The complex optimization was performed using particle swarm optimization, (which will be explained in Section 3) as developed by Kennedy and Eberhart (Kennedy and Eberhart [20]; Eberhart et al. [21]).

## 2. Material and Methods

### 2.1. Darvas Box

In this research, we programed different Darvas box and Linear Regression setups for Bitcoin– U.S. dollar daily data from the beginning of 2012 until the end of March 2020. (The date consists of open, close, upper price and lower daily price.) This period was split to three years of training periods (SVM training period) and more than five years for testing the optimal setups. Figure 1 shows an example for the automated trading platform using Darvas Box.

Figure 1 show how Darvas boxes are designed and when applied to Bitcoin’s data, they generate long and short signals. It is important to note that the trade signals symbolize a forecasted beginning of a price trend (up or down) based on price breakout of the box designed with predetermined number of days. The example presented in Figure 1 demonstrates an upward breakout, which signals the beginning of a long trend and a down breakout, which symbolizes the beginning of a downtrend. The long trend noted in Figure 1 continued for 50 days before a downtrend took place. This algorithmic trading system assumes that the trader is always exposed to price shifts between long and short positions.

### 2.2. Conditional Entropy and the Chain Rule

In information theory, the conditional entropy quantifies the amount of information needed to describe the outcome of a random variable given that the value of another random variable is known. The entropy of Y conditioned on X is written as H(Y⋮X). The chain rule assumes that the combined system determined by two random variables X and Y has joint entropy H(X,Y). Now, if we first learn the value of X, we have gained H(X) information. Once X is known, we only need H(X,Y)−H(X) to describe the state of the whole system. This quantity is exactly H(Y⋮X), which gives the chain rule of conditional entropy [22].
(1)H(Y⋮X)=H(X,Y)−H(X)

In this research, we test whether Darvas box signaling can serve as a chain conditional entropy for Bitcoin’s future price changes. The proposition behind the Darvas box trading strategy opposes the famous “Random Walk” hypothesis that argues that historic price changes are unable to predict future price changes is given by Equation (2).
(2)$$\left(HU_{t}\vdots DU_{t-1}\right)=HU_{t}$$
where $HU_{t}\vdots DU_{t-1}$ is the probability for a price change up at day *t* given that a Darvas up signal appeared at day *t* − 1. $HU_{t}$ is the probability for a price change up at day *t* at a non-Darvas box upsignaling at the previous day.

The “Random Walk” hypothesis is consistent with the “Efficient Market” hypothesis in their statements that future price changes are impossible to be predicted using past information. In contrast of that, if the Darvas box previous day signaling can predict the current day price change we should expect that the probability for the current day uptrend would be higher when a Darvas box up signal (The same analysis can be done for Darvas box downtrend signaling) appeared a day before and that the signaling effect will fade over time (Equations (3) and (4)).
(3)$$\left(HU_{t}\vdots DU_{t-1}\right)>HU_{t}$$
(4)$$\left(HU_{t}\vdots DU_{t-1}\right)>\left(HU_{t}\vdots DU_{t-2}\right)>\left(HU_{t}\vdots DU_{t-n}\right)$$
where $HU_{t}\vdots DU_{t-n}$ is the probability for a price change up at day *t* given that a Darvas up signal appeared at day *t* − *n*. $HU_{t}$ is the probability for a price change up at day *t* at a non-Darvas box up signaling at the previous day.

### 2.3. Linear Regression

Figure 2 demonstrates how we use the Linear Regressions technique for trading Bitcoin.

When designing and deploying a Linear Regression strategy, one must first determine the length of time for the constructing of the regression line. Second, the designer must decide about the span from that line that determine the entry and exit from the trading positions. The regression line in Figure 2, for example, is based on 50 trading days when one standard deviation from that line determine the entry and exit points to the trading position. When the actual price drift upward from the +1 standard deviation line the system enters a long position that ends when the price cross down the regression line. A short position begins when the Bitcoin price cross down the −1 standard deviation line. For the initial analysis of the capabilities of the Linear Regression to predict future Bitcoin’s prices we run autoregression Equation (5).
(5)$$Bp_{t}=\beta_{1}+\beta_{2}Bp_{t-1}+\beta_{3}Bp_{t-2}+\ldots \beta_{8}Bp_{t-7}$$
where $Bp_{t}$ = Bitcoin daily price change at day *t*, $Bp_{t-n}$ = Bitcoin daily price change at day *t* − *n* (*n* = 1–7).

### 2.4. Particle Swarm Optimization

As described above, the designation of both systems (Darvas box and Linear Regression) need multi-objective optimization formulas. In many real-life problems, objectives under consideration conflict with each other, and optimizing a solution with respect to a single objective can result in unacceptable results with respect to the other objectives. A reasonable solution to a multi-objective problem is to investigate a set of solutions, each of which satisfies the objectives at an acceptable level without being dominated by any other solution. Many of these processes were developed over the years and are used to find solutions to various complex problems. We selected particle swarm optimization that was developed by Kennedy and Eberhart (Kennedy and Eberhart [20]; Eberhart et al. [21]) as our primary optimization method. Eberhart and Shi [23] demonstrated that it could be successfully applied to tracking and optimizing dynamic systems for most optimization problems. However, the most promising applications of this process are in robotics, decision making and simulations, all of which are related to our mission. The particle swarm optimization involves using a stepwise process to change the velocity of each particle toward its best location that maximizes our target function. Next, for each set of particles, we evaluated the desired optimization fitness function using our predefined goals: Minimum Maximum Drawdown (MDD) and maximum Percentage of Profitable trades (PP), Profit Factor (PF) and Net Profit (NP) (the optimization function will be explained later). We then compared the fitness of the setups with “*pbest*.” (“*pbest*” = The setup that achieved the best results in reducing the maximum drawdown and maximizing the percentage of profitable trades, the profit factor, and the net profit). If the current value was better than “*pbest*”, we set the “*pbest*” value to the current value. In addition, we compared the evaluations of the fitness of the setups with the population’s overall previous best. If the current value was better than “*gbest*,” (“*gbest*” = global best identification), we reset “*gbest*” to the current value and setups. Finally, we changed the trading setups according to Equation (6) and calculated the results:*V* (1) *i*+1,*d* = *Vid* + *C*1*Rand* ∗ _*Pid* − *Xid*_ + *C*2*Rand* ∗ _*Pgd* − *Xid*(6)
*X* (2) *i*+1,*d* = *Xid* + *Vid*(7)
where *Vid* = the value of each particle of the setup, *Rand* = random number, *Pid* = the particle’s initial identification and *Pgd* = the particle’s global best identification.

We then looped to step 2 (Equation (7)) until the best results were achieved. The use of a random variable in the above process is essential for keeping the optimization process unbiased and to ensure that all variables have an equal chance of entering the process. The random variable represents an initial setup for the trading systems. For example, the Linear Regression system needs two main inputs: number of days and number of standard deviations. The initial random setup is generated into the trading system and the predefined trading results are calculated (MDD, PP, PF and NP). In the next steps, the optimization process alters the setups to achieve better predefined trading result.

As mentioned above, our trading system was designed to report the actual algorithmic trading results: MDD, PP, PF and NP. MDD calculation is used to assess the relative riskiness of one trading strategy versus another, as it focuses on capital preservation, which is a key concern for most risk-averse investors. It measures the largest decline in the value of a portfolio before a new peak is achieved. MDD assesses only the size of the largest loss, without taking into consideration its frequency or how long it will take an investor to recover from that loss. A low maximum drawdown is preferable because it indicates that losses from the investment are small. The MDD is provided in both dollar terms and as the percentage of the amount invested. The PP provides information about the percentage of profitable trades in relation to all trades. If it is above 50%, the trading system has generated more winning trades than losing trades. However, this does not mean that the net profit of all trades is positive and vice versa. A score less than 50% does not mean that the trading system is losing money. The profit factor (PF) is defined as gross profits divided by gross losses. The result indicates the difference between the system’s gains and losses. For example, if the profit factor is equal to 1.2, the system generated 20% more profits than losses. Net profit (NP) is the dollar value net profit generated by the trading system. Although one might assume that the three profit indicators (PP, PF and NP) move together, in fact they can vary dramatically, and therefore may confuse investors and algorithmic trading planners. Each individual investor picks at least one parameter as his/her target function according to the investor risk averse level. For example: a more risk averse investor will probably choose to minimize MDD rather than maximize profits while a risk seeker investor will prefer maximizing NP or PP as his primary target.

## 3. Results

### 3.1. Darvas Box

For the total examined period, the average daily Bitcoin price change was 0.34% with standard deviation of 0.045. Table 1 summarizes the results of four days price change after an up/down Darvas box signaling.

Table 1 contradicts the “Efficient Market” hypothesis and prove that Darvas box signaling do predict short term price changes. The table also show that the Darvas signaling works better predicting an uptrend than a downtrend and that the signal fades over time. The average four days gains after an up signal was 0.62% which is almost double than the entire period average daily return (0.34%). The four days average return after a down signal was −0.05% when on average the short-term signal effect faded after two days.

We now start to examine a long-run simulated trading strategy based on Darvas box signaling, through our trading platform (Tradingview.com). The strategy is quite simple: investors change their Long/Short positions only if a new opposite signal appears. The optimization process was conducted according to Equations (6) and (7) and the results of the testing period are summarized in Table 2.

Table 2 demonstrates that all the tested Darvas box setups have achieved a high and above 1 profit factors, meaning that the trading strategies have generated high gains for all examined days. The best setup was found to be six days. Six days of formation has gained 23.84% net profit with 55.63% winning trades. Six days of formation has also produced the least risky setup with MDD of 1.67%. The results summarized in Table 2 prove that Darvas boxes are useful in forecasting Bitcoin’s trends. For the entire testing period (the training period began at 1.1.2012 until 31.12.2014, and the strategy testing period began at 1.1.2015 until 30.3.2020), the Bitcoin price rose from $314 to $6407 for a profit of $6093 (1940%) for a single Bitcoin. In comparison, the six days of Darvas box trading strategy resulted in a $23,835 profit (7590%). The worst strategy described in Table 2 (5 days of setup) resulted in a profit of $9675 (3081%), which is 51% better than the buy-and-hold result. Table 3 summarizes the trading results of six days of Darvas formation forecasts divided to long and short trades.

Table 3 shows that although the profit factor is high above 1 for both trends, the technique is more useful for long trends rather than short trends (Darvas himself only used his techniques for long trades). For a long trend, the PP is 61.29% with PF of 6.4. It is worth noting that the average forecasted trend is 50 days, meaning that a Darvas box breakout can forecast Bitcoin price directions for 50 days.

### 3.2. Linear Regression

We start our analysis by running an auto regression model that predicts the Bitcoin daily price change at day *t* ($Bp_{t})$, by price changes at days *t* − *n* ($Bp_{t-n}$, *n* = 1.7) (Table 4).

The autoregression model summarized in Table 4 shows that the daily Bitcoin’s price change is significantly negatively influenced by past changes of days *n* = −1 and *n* = −2 (For *n* = −1 the significant level *p* = 10% and for *n* = −2 the significance level is *p* = 5%), positively influenced by days *n* = −4 and *n* = −5 (For both *n* = −4 and *n* = −5, the significance level is *p* = 5%) and then again negatively by days *n* = −6 and *n* = −7 (For *n* = −6 the significant level *p* = 10% and for *n* = −7 the significance level is *p* = 5%). Those results suggest two days of price cyclicality. We use the particle swarm optimization process for our SVM training period, optimizing the various Linear Regression strategy particles and testing the best-found setups for the remaining examined time frame (the SVM training period was three years, while the testing period was more than five years). Table 5 summarizes the results of the algorithmic trading using the Linear Regression Technique.

Table 5 shows that the best setup for Linear Regression base algorithmic trading is setup number 5 (42 days with 1 standard deviation), which resulted in 3.88 PF, 53.68% of winning trades and $29,667 NP, which is 4.86 times the buy-and-hold profit for the same period of trading time (from 1.1.2015 until 30.3.2020). That setup has also generated the lowest MDD (2.22% (other setups have also achieved that MDD result)) meaning that it is relatively safe trading strategy. Second, it was setup number 9 (44 days with 1 standard deviation) that resulted in 3.49 PF, 53.44% of winning trades and $28,601 NP. The worst setup generated by our system (setup number 16) resulted in a profit of $13,396, which is 2.19 times the buy-and-hold profit for the entire period of trading time. Splitting the two best setups into long and short trades have resulted in the following, as shown in Table 6.

Table 6 indicates that for both best setups, long trading were more profitable than short trading in terms of the PF and PP, indicating that the Linear Regression model forecast better uptrends than downtrends. Moreover, the average number of days in the trade is longer for the long trades than for the short trades (20 and 21 days opposed to 16 days.).

## 4. Discussion

Previous Bitcoin studies have concentrated mainly on the link between Bitcoin’s price and social networks like Google and Twitter. Others have analyzed Bitcoin’s market anomalies. More recent papers have acknowledged the contribution of complex algorithmic machine learning to Bitcoin trend forecasting. We contribute to recent scholarly studies by testing the effectiveness of Darvas box and Linear Regression techniques in predicting Bitcoin’s price trends. In our study, we used a supervised learning technique that enables a machine to learn a function from training data that consists a set of examples and implement the knowledge it aggregates to predict future events. In information theory, the conditional entropy quantifies the amount of information needed to describe the outcome of a random variable given that the value of another random variable is known. In this research, we test whether Darvas box and Linear Regression signaling can serve as a chain conditional entropy for Bitcoin’s future price trends. The proposition behind the Darvas box trading strategy opposes the famous “Random Walk” and “Efficient Market” hypotheses that argue that historic price changes are unable to predict future price changes. Since the designation of both systems demands a multi-objective optimization formula, we selected particle swarm optimization as our primary optimization method. The particle swarm optimization involves using a stepwise process to change the velocity of each particle toward its best location that maximizes our target function. For each set of particles, we evaluated the desired optimization fitness function using our predefined goals: MDD, PP, PF and NP. Moreover, each individual investor picks at least one parameter as his/her target function according to the investor risk-averse level. Our research is based on Bitcoin–U.S dollar prices from the beginning of 2012 until the end of March 2020. That period was split to three years of a training period and more than five years of testing. Results show that the Bitcoin price change does not follow the “Random Walk” and the “Efficient Market” hypothesis and they can be predicted using the two examined methodologies. Both methodologies work better to predict uptrends than downtrends. The best setup for the Darvas box strategy is a six-day formation breakout. A Darvas box uptrend signal was found to able to predict four sequential daily returns while a down signal faded after two days on average. The best setup for the Linear Regression model was found to be 42 days with 1 standard deviation. At the tested period, the Bitcoin’s price has risen from $314 to $6407 yielding a profit of $6093. In comparison, the best Darvas box trading setup has resulted in a $23,835 net profit, which is 3.9 times the buy-and-hold strategy. The best Linear Regression setup resulted in $29,667 net profit, which is 4.86 times the buy-and-hold profit for the same period of trading time. Finally, our autoregressive model has proven that past daily Bitcoin price shifts significantly impact current price shifts with two days in a row of inverting signs.

## Figures and Tables

**Figure 1 entropy-22-00838-f001:**
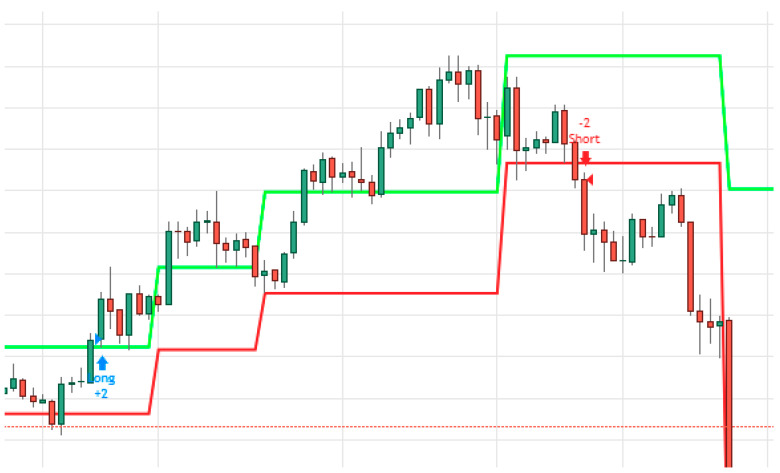
Darvas Box Trading Strategy. Notes: Every candlestick in Figure 1 represents the high/low, open/close Bitcoin daily prices. The green and red lines indicate the upper and lower boundaries of Darvas’ Boxes.

**Figure 2 entropy-22-00838-f002:**
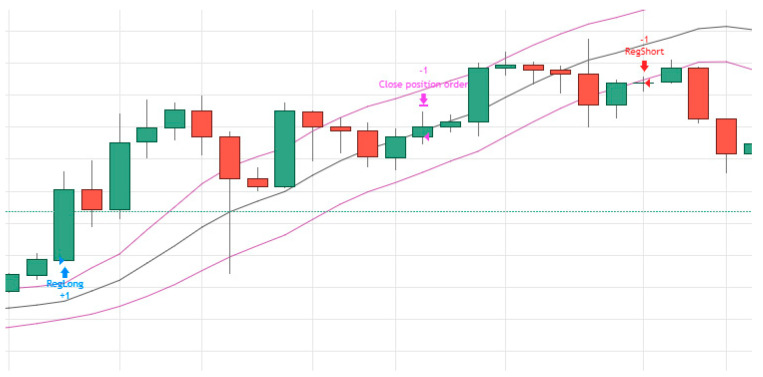
Linear Regression Trading Strategy. Notes: Every candlestick in the Figure 1 represent the high/low open/close Bitcoin daily prices. The middle line in Figure 2 represent the Linear regression line while the other two lines represent one standard deviation from the linear line.

**Table 1 entropy-22-00838-t001:** Bitcoin Price Change after Darvas Box Signaling.

Signaling Date.	Darvas Up	Darvas Down
*t* − 1	1.21% (0.025)	−0.62% (0.031)
*t* − 2	0.75% (0.038)	−0.12% (0.056)
*t* − 3	0.24% (0.046)	0.15% (0.052)
*t* − 4	0.32% (0.053)	0.25% (0.061)
average	0.62% (0.04)	−0.05% (0.05)

**Table 2 entropy-22-00838-t002:** Darvas Box Trading Results.

Box Number of Days	PF	PP	NP	MDD	Number of Trades
10	3.67	50	16,785 (16.80)	2478 (2.25)	40
9	3.19	52.27	15,362 (15.36)	2478 (2.13)	44
8	3.50	52.08	18,197 (18.20)	2478 (2.08)	48
7	3.19	52.73	16,762 (16.76)	2478 (2.14)	55
6	4.82	55.63	23,835 (23.84)	2032 (1.67)	63
5	1.45	47	9675 (9.68)	6625 (6.20)	100

Notes: PF = profit Factor, PP = Percentage of Profitable trades, NP = Net Profit, the brackets below hold the percentage term. MDD = Maximum Draw Down, the brackets below hold the percentage term.

**Table 3 entropy-22-00838-t003:** Six days of detailed long/short trading results.

	PF	PP	NP	Average Days in Trade	Number of Trades
Long	6.40	61.29	15,583	46	31
Short	3.44	50.00	8252	54	32
All	4.82	55.63	23,835	50	63

Notes: PF = profit Factor, PP = Percentage of Profitable trades, NP = Net Profit.

**Table 4 entropy-22-00838-t004:** Bitcoin’s auto regression model.

	α	$Bp_{t-1}$	$Bp_{t-2}$	$Bp_{t-3}$	$Bp_{t-4}$	$Bp_{t-5}$	$Bp_{t-6}$	$Bp_{t-7}$
Coefficient (t Stat)	0.003* (2.72)	−0.032 (−1.83)	−0.115 * (−5.81)	0.017 (1.40)	0.227 * (12.6)	0.073 * (3.87)	−0.031 (−1.73)	−0.035 (−1.96)
			*n* = 3013	$R^{2}=0.077$	F = 50.57			

Notes: $Bp_{t-n}$ = Bitcoin daily price change at day t-*n* (*n* = 1–7), * = Statistically significant.

**Table 5 entropy-22-00838-t005:** Linear regression trading results.

Setup	No. of Days	Standard Deviation	PF	PP	NP	MDD	Number of Trades
1	40	1	3.24	55.92	26,720 (26.72)	2794 (2.65)	152
2		2	2.35	51.49	12,885 (12.89)	2657 (2.32)	101
3	41	1	3.36	55.24	27,635 (27.64)	2622 (2.49)	143
4		2	2.29	52.04	13,416 (13.42)	3162 (2.91)	98
5	42	1	3.88	53.68	29,667 (29.66)	2377 (2.22)	136
6		2	2.42	52.63	13,508 (13.51)	2912 (2.69)	95
7	43	1	3.19	52.21	26,345 (26.35)	2377 (2.22)	136
8		2	2.30	56.04	12,870 (12.87)	3347 (2.99)	91
9	44	1	3.49	53.44	28,601 (28.60)	2377 (2.22)	131
10		2	3.41	58.24	18,079 (18.08)	3460 (2.95)	91
11	45	1	2.94	54.69	24,505 (24.51)	2398 (2.22)	128
12		2	3.27	58.62	16,884 (16.88)	3367 (2.90)	87
13	50	1	2.75	56.03	20,191 (20.19)	4478 (3.82)	116
14		2	2.90	56.79	16,502 (16.50)	4433 (3.84)	81
15	55	1	2.58	50.46	18,411 (18.41)	5053 (4.28)	109
16		2	2.30	50.67	13,396 (13.40)	5421 (4.74)	75

Notes: PF = profit Factor, PP = Percentage of Profitable trades, NP = Net Profit, the brackets below hold the percentage term. MDD = Maximum Draw Down, the brackets below hold the percentage term.

**Table 6 entropy-22-00838-t006:** The best two setups long/short trading results.

Setup		PF	PP	NP	Average Days in Trade	Number of Trades
5	Long	6.62	66.18	19,200	20	68
	Short	2.52	41.18	10,467	16	68
	All	3.88	53.68	29,667	18	136
9	Long	5.42	68.25	17,181	21	63
	Short	2.50	39.71	11,420	16	68
	All	3.49	53.44	28,601	19	131

Notes: PF = profit Factor, PP = Percentage of Profitable trades, NP = Net Profit.

## Data Availability

The data of the Bitcoin’s prices can be provided upon request. Moreover, all programming script will be provided upon request.

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
