# Peer review of "Forecasting Bitcoin Trends Using Algorithmic Learning Systems"

_entropy, 2020, doi:10.3390/e22080838_

Round 1
Reviewer 1 Report
For the data availability statement I suggest creating and sharing via manuscript Google Colab Notebook. I would like no know how your method compared to simply buy and hold strategy. If I bought 1 BTC in 2012 would I end up with more money that using prediction of your method or with less and why? Author do not discuss or compare their results to any state of the art works, instead they refer to in average 5-6 y old literature and even older methods. By doing do so the results are not trustworthy. I would like to see in results comparison with with current crupto singals recommendations or bots and scientific work like:
Wołk, K. (2020). Advanced social media sentiment analysis for short‐term cryptocurrency price prediction. Expert Systems, 37(2), e12493.
Abraham, J., Higdon, D., Nelson, J., & Ibarra, J. (2018). Cryptocurrency price prediction using tweet volumes and sentiment analysis. SMU Data Science Review, 1(3), 1.
Jay, P., Kalariya, V., Parmar, P., Tanwar, S., Kumar, N., & Alazab, M. (2020). Stochastic Neural Networks for Cryptocurrency Price Prediction. IEEE Access, 8, 82804-82818.
And more….
Reviewer 2 Report
Entropy-869650-Review Report
First, I think the author(s) for an interesting and novel study on bitcoin price trend forecasting. I have a few suggestions to be considered for further improvement of the study before publication in Entropy.
- Please break the introduction section in a few paragraphs. Having paragraphs rather than very long texts improves readability. For instance, the authors can create paragraphs in lines- 47 (The second group…), 62 (As described above…), 76 (The pattened…) and 88 (Since the introduction…).
- The authors should include at least couple of existing bitcoin price forecasting studies in introduction section. For example, (1) Munim, Z. H., Shakil, M. H., & Alon, I. (2019). Next-day bitcoin price forecast. Journal of Risk and Financial Management, 12(2), 103. (2) Atsalakis, G. S., Atsalaki, I. G., Pasiouras, F., & Zopounidis, C. (2019). Bitcoin price forecasting with neuro-fuzzy techniques. European Journal of Operational Research, 276(2), 770-780.
- What do you mean by the footnote 4? The authors mention optimization function will be explained later, but they present it just after couple of sentences. I think there is no need for footnote 4.
- In the results section, please mention which software or analysis tools were used.
- Please present the results of linear regression in a Table format, which will improve readability.
- It is unclear how the particle swarm optimization was implemented. Please explain the implementation in greater detail.
- The authors often used the word “complex” for refer to the methods used. Please remove this word as it does not add much value.
- Although the manuscript is well written, I recommend professional language proofreading as some minor language issues are found throughout the manuscript.
Good luck with the revision.
Reviewer 3 Report
Please see review report.
This is a very interesting piece that uses particle swarm optimization to find the best forecasting combinations involving bitcoin. The results show that that the bitcoin price changes do not follow the "Random Walk" efficient market hypothesis and shows that both Darvas Box and Linear Regression can help traders to predict the bitcoin's price trends.
The empirical implementation appears appropriate and I believe that this piece as the potential to garner readership and future studies based off its methodology and reported findings.
Reviewer 4 Report
The paper language needs to be entirely professionally proof-read. I wanted to highlight some specific parts that need to be improved. But the whole paper needs to be checked.
In addition, the findings' implications need to be discussed properly. As of now the paper merely presents the findings. No practical and theoretical implications are discussed. This minimizes the contribution and significance of the study.
Round 2
Reviewer 1 Report
The author improved article in accordance to my comments but misunderstood some parts. I provided 3 papers not to be included but as an examples that could be included if authors found them important, but it was not my task to determine which to include and which not to include. Authors should do more deep literature review not just copy my examples which were in fact very limited. In summary I am trying to say that including that references is good but should be more deeply analyzed and justified with other similar articles on the topic. Luckily author added on his own also two more recent papers citations: (1) Munim, Z. H., Shakil, M. H., & Alon, I. (2019). Next-day bitcoin price forecast. Journal of Risk and Financial Management, 12(2), 103. (2) Atsalakis, G. S., Atsalaki, I. G., Pasiouras, F., & Zopounidis, C. (2019). Bitcoin price forecasting with neuro-fuzzy techniques. European Journal of Operational Research, 276(2), 770-780 and proven that his method outperforms buy hold strategy. Discussion includes all this and has been greatly included. I believe that even if publication is not yet perfect it is marginally tend to publication, which is my recommendations.
Reviewer 2 Report
Thank you for addressing my suggestions. I believe the manuscript is now publishable in Entropy.
Reviewer 4 Report
Great Job. Well done.
This manuscript is a resubmission of an earlier submission. The following is a list of the peer review reports and author responses from that submission.